# Oxidative Damage to Various Root and Shoot Tissues of Durum and Soft Wheat Seedlings during Salinity

**Neonila Kononenko** [1,*] **, Ekaterina Baranova** [1] **, Tatyana Dilovarova** [1] **, Eduard Akanov** [2] **and Larisa Fedoreyeva** [1,3]

1   All-Russia Research Institute of Agricultural Biotechnology, Timiryazevskaya 42, 127550 Moscow, Russia; greenpro2007@rambler.ru (E.B.); dilovarova@yandex.ru (T.D.)
2   D.N. Pryanishnikova All-Russian Research Institute of Agricultural Chemistry, Pryanishnikov St. 31a, 127434 Moscow, Russia; en_akanov@mail.ru
3   A.N. Belozersky Institute of Physico-Chemical Biology, M.V. Lomonosov Moscow State University, Leninskie Gory 1, building 40, 119991 Moscow, Russia; fedlara@inbox.ru
*   Correspondence: nilava@mail.ru; Tel.: +7-903-7689802

**Abstract:** The toxicity of high concentrations of sodium chloride creates significant difficulties in realizing the productivity potential of wheat. The development of effective test systems for the identification and selection of resistant genotypes is an urgent task given the global increase in soil salinity in agricultural land. To identify the characteristics of the plant's reaction to the toxic effect of sodium chloride, wheat genotypes with different resistance to ionic toxicity (the Orenburgskaya 10 and Orenburgskaya 22 varieties) were used. In model experiments, we used fluorescence, light-optical and electron microscopy to characterize the structural and functional features of the cells of the roots of wheat seedlings, and cytological markers suitable for creating a test system for the early diagnosis of the sensitivity of wheat genotypes to sodium chloride were established. The response of the plants to the effects of sodium chloride was assessed by changes in biometric data, respiration rate, peculiarities in the accumulation of reactive oxygen species (ROS) and mitochondrial staining, and the quantitative assessment of coleoptile cell viability as putative sensitivity markers. In the sodium chloride-sensitive genotype (Orenburgskaya 10), toxic effects resulted in oxidative damage in the root cells, while in the resistant genotype (Orenburgskaya 22), oxidative damage to the cells was minimal. A high level of expression of mitochondrial superoxide dismutase (*MnSOD*) was found in the roots of the Orenburgskaya 22 variety. The identification and functional analysis of cytological and molecular markers provide the basis for further studies of the resistance of wheat to sodium chloride stress.

**Keywords:** NaCl; mitochondria; salt tolerance; morphology; *Triticum durum* Desf.; *Triticum aevistium* host.; ROS; MnSOD

## 1. Introduction

Under natural conditions, plants are frequently subject to various stresses, such as drought, salinization, extreme temperatures, and the presence of heavy metals in the environment, which can seriously affect their growth and development [1]. Excessive salinization of the soil is unfavorable for most cultivated plants. Salts affect the physiological, biochemical and molecular functions of plants, and consequently, lead to a decrease in productivity and the quality of crops around the world [2]. Salinization of agricultural land tends to increase steadily as a result of secondary salinization processes, which limits the use of agricultural land in arid areas.

Stress causes the formation of reactive oxygen species (ROS), including superoxide anion radicals ($O_2^-$), hydrogen peroxide ($H_2O_2$), hydroxyl radical ($OH^-$), peroxyl radicals ($HO^-$), and singlet

oxygen ($^1O_2$). As a result of peroxidation and destruction of macromolecules, cell membranes are damaged, which ultimately leads to cell death [3,4]. In plant cells, ROS production is strictly controlled by enzymatic and non-enzymatic antioxidant defense systems, including superoxide dismutase (SOD), catalase, ascorbate peroxidase, monodehydroascorbate reductase, dehydroascorbate reductase, thioredoxin, and glutathione. Among antioxidant enzymes, SOD can serve as an effective absorber of ROS, catalyzing the decomposition of anion radicals of superoxide ($O_2^-$) to hydrogen peroxide ($H_2O_2$), which is subsequently converted to non-toxic water and oxygen.

Depending on the metal interacting with the active site, enzymes can be divided into four types: FeSOD, MnSOD, Cu/ZnSOD and NiSOD [5–7]. Different SOD isoforms with similar functions have different metal cofactors, amino acid sequences, crystal structures, and subcellular localizations and exhibit different sensitivity to $H_2O_2$ in vitro [8]. For example, KCN and $H_2O_2$ can irreversibly inactivate Cu/ZnSOD and FeSOD, but MnSOD is not sensitive to any of the chemical substances [9]. Cu/ZnSOD, which are mainly located in chloroplasts, cytoplasm and/or extracellular spaces, are present in some bacteria and all eukaryotic species [10], while MnSODs are mainly found in plant mitochondria [11]. At least one copy of MnSOD present in plant genomes plays a role in the removal of ROS in mitochondria [12]. FeSODs are common in prokaryotes, protozoa, chloroplasts, and plant cytoplasm [13], while NiSODs are present in Streptomyces and were predicted in some cyanobacteria [14], however, they have not been found in plants.

Active forms of oxygen, such as superoxide, $H_2O_2$, and hydroxyl radicals, are by-products of normal cellular metabolism. These reactive oxygen species lead to lipid peroxidation of membranes [15], damage to DNA chains [16] and the inactivation of enzymes [17]. The conditions leading to damage caused by reactive oxygen species are called oxidative stress. Both chloroplasts and mitochondria can produce ROS, either under normal growth conditions or during exposure to various stresses. The PSI electron transport chain contains a number of auto-oxidable enzymes that reduce $O_2$ to superoxide [18], and evidence suggests that superoxide and $H_2O_2$ can also be produced by PSII under high light intensity [19]. During mitochondrial respiration, ROS are also generated via electron transport chain reactions [20].

The aim of this study was to establish an early (at the seedling stage) differential diagnosis of oxidative damage to soft and durum wheat tissues when exposed to high concentrations of sodium chloride to identify phenotypic and cytological sensitivity targets.

## 2. Material and Method

### 2.1. Object of Study

This study included a variety of soft wheat (*Triticum aestivum* Host.), the spring form Orenburgskaya 22, and a spring form of durum wheat (*Triticum durum* Desf.), Orenburgskaya 10, obtained from the collection of the Federal State Budget Scientific Institution of the Federal Scientific Center of the Russian Academy of Sciences, Orenburg, Russia.

Sensitivity to chloride salinity was investigated using the roll culture method [21]. The seeds of two varieties of wheat (50 seeds) were laid out on paper rolls. The paper rolls (4 pieces each) were placed in glasses, and water or 150 mM NaCl was added to 150 ml of each solution. The experiment was carried out in 3 replicates (3 glasses each). Glasses were placed in a climatic chamber. Cultivation was carried out at 24 °C with artificial lighting by daylight lamps (5000 lux) and a day/night cycle of 10/14 h, respectively. After 7 days of cultivation, half of the rolls were transferred to other conditions: rolls with water were transferred to 150 mM NaCl, and rolls from 150 mM NaCl to water. The cultivation of all rolls continued for another 3 days, according to the scheme shown in Figure 1. After 10 days of cultivation the weight of wet and dry biomass, the length of the main root, and the height of the shoot were measured. The program Statistica 5.0, Student's parametric criteria, and standard Microsoft Excel software were used for date evaluation The mean values ($n = 30$) and their standard deviations are shown according to Student's criterion, $p < 0.05$.

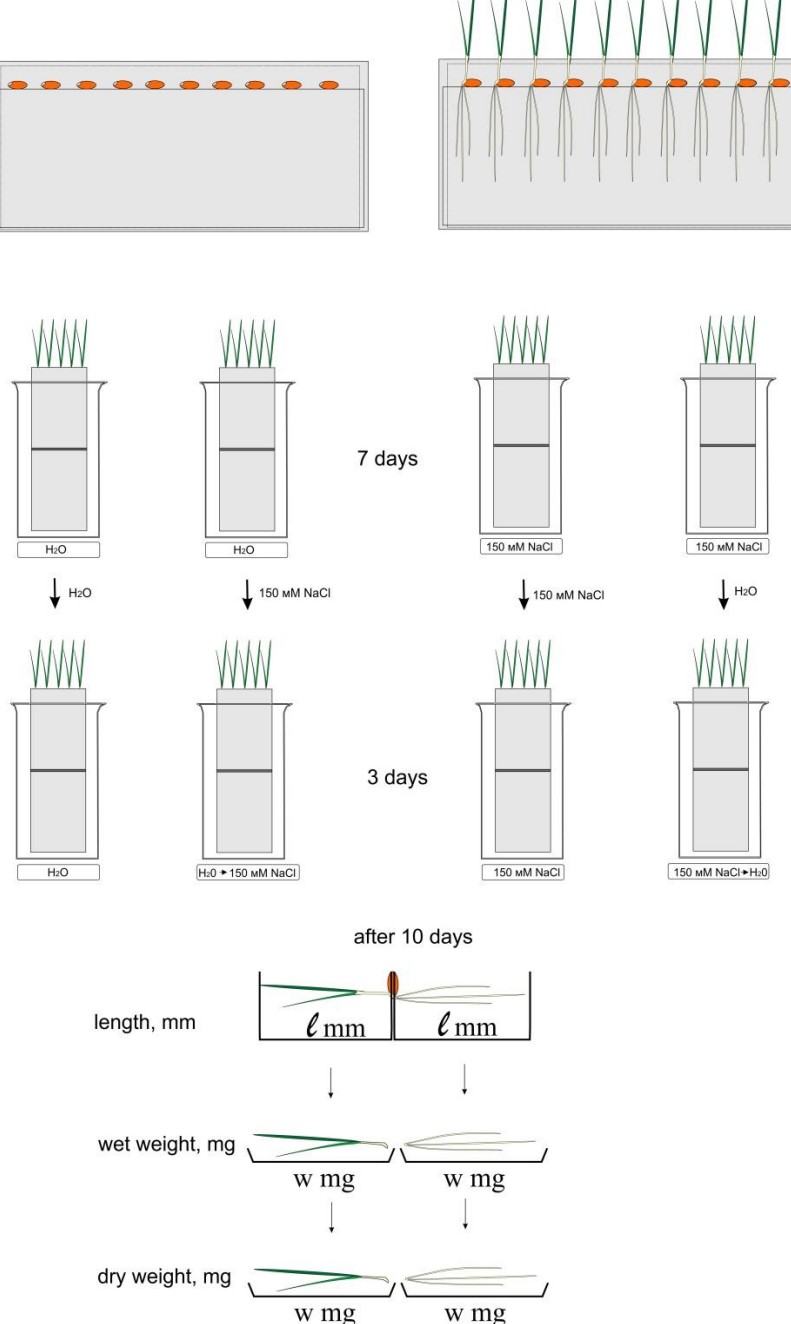

**Figure 1.** Experimental scheme. Paper rolls were cultivated in distilled water and 150 mM NaCl. After 7 days of cultivation, half of the rolls were transferred to other conditions: rolls with water were transferred to 150 mM NaCl, and rolls from 150 mM NaCl to water. The cultivation of all rolls continued for another 3 days.

### 2.2. Staining with Trypan Blue

At day 10, intravital staining of coleoptile with trypan blue (AppliChem GmbH, Darmstadt, Germany)-0.5% aqueous solution was carried out for 5 min and then washed 3 times. Visualization was carried out using light microscopy (Olympus BX51 microscope (Olympus corporation, Tokyo, Japan), ×10 lens. Images were obtained using a Color View II digital camera (Soft Imaging System, Munster, Germany).

*2.3. Fluorescence Microscopy*

The root tip (4–5 mm) of all 10-day-old wheat varieties were separated and placed on a glass slide in a drop of water. For intravital imaging in ROS cells, we used an aqueous solution of Carboxy-H2DFFDA (Thermo Fisher Scientific, Waltham, Massachusetts, USA), at a concentration of 25–50 nM, the incubation time was 30 min, followed by 3 washes in distilled water.

For intravital imaging of mitochondria in root cells, MitoTracker green dye (MitoTracker Green FM, Thermo Fisher Scientific, Waltham, Massachusetts, USA) was used at a concentration of 20–40 nM, the incubation time was 30 min, followed by 3 washes in distilled water. Then, the live roots were placed in experimental solutions without stain. Intravital preparations were analyzed using an Olympus BX51 fluorescence microscope (Olympus corporation, Tokyo, Japan), with magnification ×10, at a wavelength of 490 nm. Images were obtained using a Color View II digital camera (Soft Imaging System, Munster, Germany). The calculation of the main statistical parameters was carried out according to standard methods, and Statistica 6.0 and STATAN programs for statistical data processing were used.

*2.4. Transmission Electron Microscopy (TEM)*

Root apex segments (4 mm) were fixed for 24 h in 2.5% glutaraldehyde (Merck, Darmstadt, Germany) dissolved in 0.1 M of Sorensen's phosphate buffer with 1.5% sucrose (pH 7.2). Then the samples were washed and post-fixed in 1% OsO4 (Sigma-Aldrich, Sant-Louis, Missouri, USA), dehydrated in ethanol of increasing concentrations (30%, 50%, 70%, 96%, and 100%) and in propylene oxide (Fluka, Munich, Germany). The samples were embedded in a mixture of Epon-812 and Araldite (Merck, Darmstadt, Germany) according to the standard procedure. For TEM, the embedded samples were sectioned using an ultramicrotome LKB-III (LKB, Bromma, Sweden), placed on formvar coated grids and stained with uranyl acetate and lead citrate. The ultrathin sections were examined and photographed with an electron microscope H-300 (Hitachi, Tokyo, Japan). The ultrastructure of mitochondria of the root parenchyma cells was studied.

*2.5. The Respiration Intensity*

The dark respiration rate (DRR) of wheat seedlings was determined under normal conditions as well as under oxidative stress, which was caused by the action of sodium chloride solutions. For this purpose, after 10 days of cultivation, 30 seedlings in the solutions in which they were grown were placed in a chamber to determine the intensity of respiration. We determined the DRR after culturing the seedlings for 7 days and then 3 days in different conditions. Because it was necessary to determine the $CO_2$ being released by an individual plant 2 h before the measurements the test glasses were placed in the dark; a constant temperature (22–23 °C) was maintained during the recording of $CO_2$ being released by the plants. Measurements were taken by a GOA4 infrared gas analyzer (Klimavtomatika, Moscow, Russia) with a 0%–0.05% $CO_2$ scale. We determined the DRR per 1 mg dry weight. We determined the DRR (μg $CO_2$/h) by the formula: DRR = [ΣVΔC/100M × 0.2] K, where ΣV is the total volume of the closed system, including the volume of the test tube (50 $cm^3$) and measuring system (50 $cm^3$); ΔC is the change in the concentration of CO2 in the total volume during expo sure, %; M is the dry weight of the plants, mg; 0.2 is exposure time, h; and K is the conversion factor of the amount of $CO_2$ from volume units ($cm^3$) to weight units (μg) reduced to normal conditions (temperature 0 °C and pressure 1 atm).

*2.6. DNA RNA Isolation*

DNA was isolated from the roots and shoots of wheat according to the Canvax protocol (Spain). RNA was isolated from individual shoots and roots according to the standard method using reagent kits for the isolation of RNA-Extran RNA Synthol (Russia). The concentration of isolated DNA and RNA preparations were determined spectrophotometrically on a NanoPhotometer IMPLEN.

*2.7. cDNA*

*cDNA* was obtained by a standard method using a set of reagents (Synthol, Russia) for reverse transcription. The concentration of *cDNA* preparations was determined spectrophotometrically on a NanoPhotometer IMPLEN

*2.8. Real-Time PCR (RT-PCR)*

PCR-RT was performed in a CFX 96 Real-Time System (Biorad) thermal cycler. Information on the primary structure of the *MnSOD* genes (AF092524.1) was taken from the NCBI database. Primers (5'-tgc-ttg-cgt-gat-ttg-tct-gat-3 'and 5'-aga-agg-tcc-cga-cag-tgg-aa-3') were synthesized by Synthol.

Samples were prepared according to the standard method using a set of reagents for RT-PCR in the presence of SybrGreen (Synthol). The RT-PCR reaction was carried out under the same conditions for all samples: 95 °C for 5 min, activation of the polymerase, then 45 cycles: 94 °C, 30 sec; 58 °C, 30 sec; 72 °C, 30 sec. Three biological replicates were used for RT-PCR. The $2^{-\Delta\Delta Ct}$ method was used for quantification.

The relative level of gene expression was calculated by a calibration curve constructed with PCR products obtained with primers for the *GaPDh* gene.

The arithmetic mean values were calculated by the formula: $M_x = \Sigma\ X_i/n$. Standard deviations: $\sigma_x = \sqrt{D_x} = \sqrt{\Sigma(X_i - M_x)^2/n-1}$, where: $D_x$ is the dispersion.

*2.9. PCR Amplification*

PCR amplification was performed in a thermal cycler (DNAEngine, Biorad). The PCR reaction was carried out with 3 μg of DNA under the same conditions as the RT-PCR. Analysis of the obtained products was analyzed by electrophoresis in 1.5% agarose.

Equipment from the Center for Collective Use of Institute of Agricultural biotechnology of RAS was used in the study.

## 3. Results

Salinization in the studied samples caused a decrease in the growth of both the root system and shoot (Figure 2).

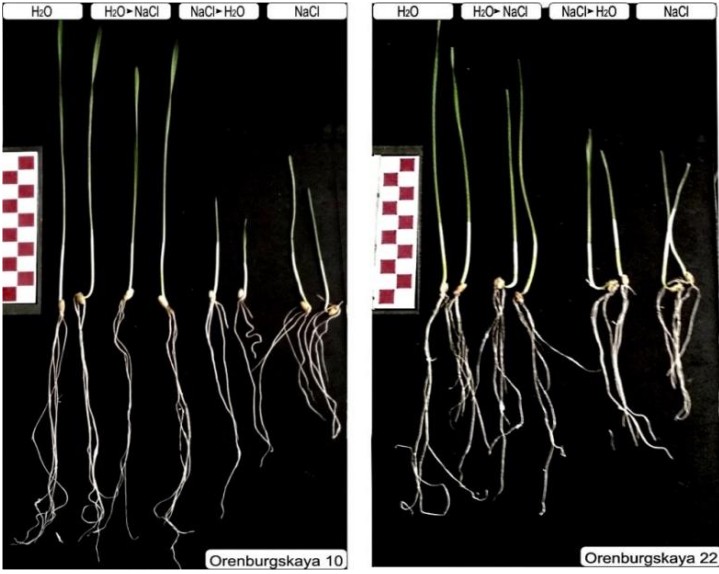

**Figure 2.** Ten-day-old seedlings of soft and durum wheat after germination under normal conditions and chloride salinization, according to the scheme. Scale ruler with 1 cm divisions.

Morphometric assessment of the seedlings allows us to evaluate the changes more clearly (Figure 3). One of the visible effects of salinization is a violation in the growth of the seedlings. Despite the varying degrees of impact on various growth processes, ultimately, the result in all cases is the inhibition of root and shoot growth., The degree to which these indicators are manifest is related to the effect of salt toxicity on plant development. As can be seen from Figure 3, the height of the aerial parts of Orenburgskaya 22 variety was more stable, so the difference in the salt-water variant compared to Orenburgskaya 10 was 25%. During salinization with sodium chloride, the difference between the varieties for the root length was about 7%.

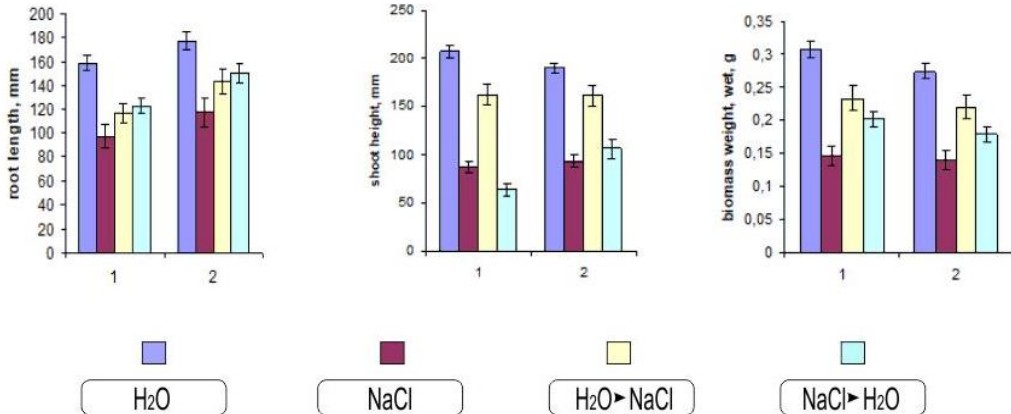

**Figure 3.** Effect of salinization on biometric characteristics of durum and soft wheat 10-day-old seedlings during salinization. 1—Orenburgskaya 10; 2—Orenburgskaya 22. The mean values ($n = 30$) and their standard deviations are shown according to Student's criterion, $p < 0.05$.

Chloride salinization can cause various morphological changes associated with changes in the habits of plants, the shape of their organs, and biomass (Figure 3).

Viability coleoptile staining with trypan blue penetrating through the membrane of dead cells was performed as a test to detect the degree of tissue damage during salinization in two varieties of wheat (Figure 4). In the control group, there was almost no visible change in coleoptile, while in the presence of sodium chloride, quite severe tissue damage was observed in some cases. So, in the Orenburgskaya 10 variety in the $H_2O + NaCl$ variant, more than 50% of cells were dead; with $NaCl + H_2O$, over 70% of cells were dead; and with NaCl, over 80% of cells were dead. At the same time, in the Orenburgskaya 22 variety in the $H_2O + NaCl$ variant, up to 20% of cells were dead; with $NaCl + H_2O$ up to 30% of cells were dead; and with saline NaCl, over 40% of cells were dead.

Thus, the most noticeable tissue damage was noted in wheat of the Orenburgskaya 10 variety. During the germination of durum and soft wheat seeds, a gradual increase in respiration occurs. In control experimental conditions, the seedlings of the studied wheat samples showed insignificant differences in respiration (Figure 5). The presence of NaCl caused a clear, acute increase in respiration, which was more obviously expressed in durum wheat. When growing conditions are changed due to an increase in salt (transferring from water to NaCl), we see that an increase (in comparison to water) but also a decrease in respiration intensity occurs (in comparison to NaCl); however, this is only significant in the case of durum wheat. When plants cultured in sodium chloride are transferred to water, there is also a significant increase in respiration.

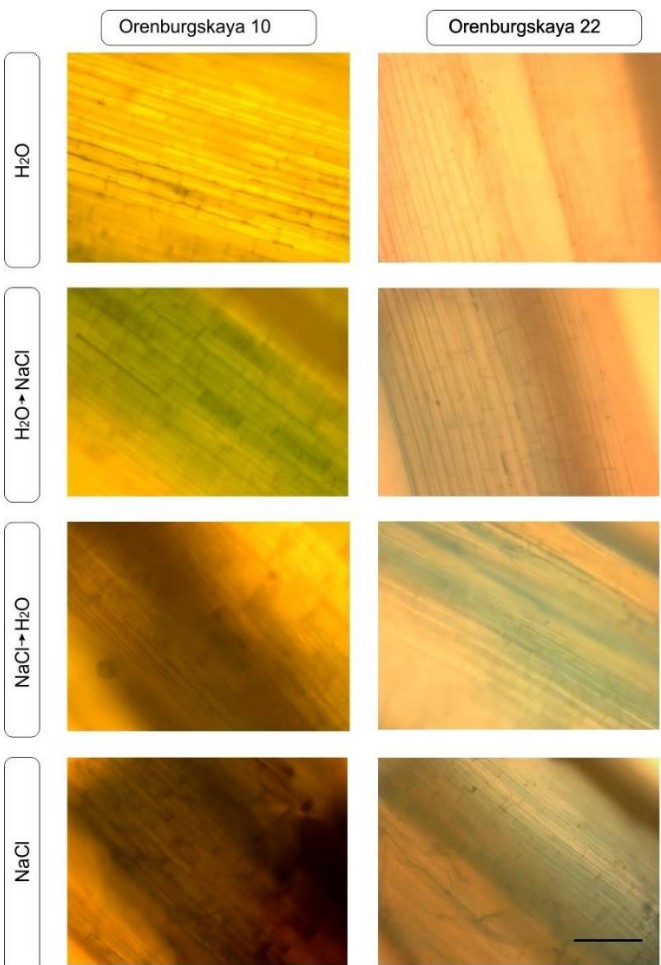

**Figure 4.** Trypan blue staining of coleoptile in 10-day-old seedlings differ in the number of dead cells. Scale segment: 400 microns.

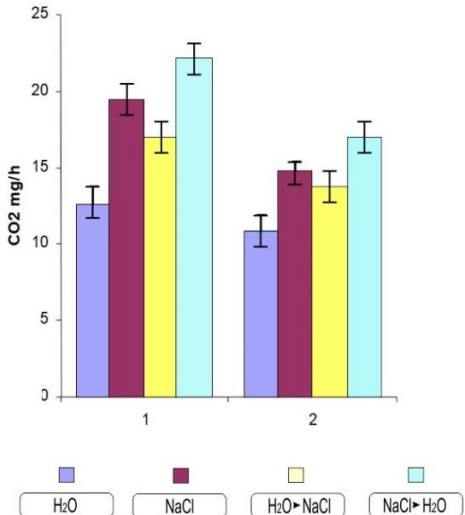

**Figure 5.** The respiration intensity of 10-day-old wheat varieties. 1—Orenburgskaya 10; 2—Orenburgskaya 22. The mean values and their standard deviations are shown according to Student's criterion, $p < 0.05$.

An important functional test for oxidative stress is the intravital staining of mitochondria using specific fluorescent markers of the MitoTracker family. We tested one of the MitoTracker variants that accumulates in both active and inactive mitochondria (stain accumulation occurs only in undamaged

mitochondria). The test showed that under the same conditions of incubation with salts, in which root cells produce an increased amount of ROS, there was a change in the nature of the mitochondrial staining (Figure 6). When cultured under normal conditions, all root cells have stained mitochondria, although the intensity of staining in different cells may vary. However, after incubation with sodium chloride, in certain areas of the root there are cells in which mitochondrial staining is completely absent, which is especially characteristic of plant roots cultivated in the presence of NaCl.

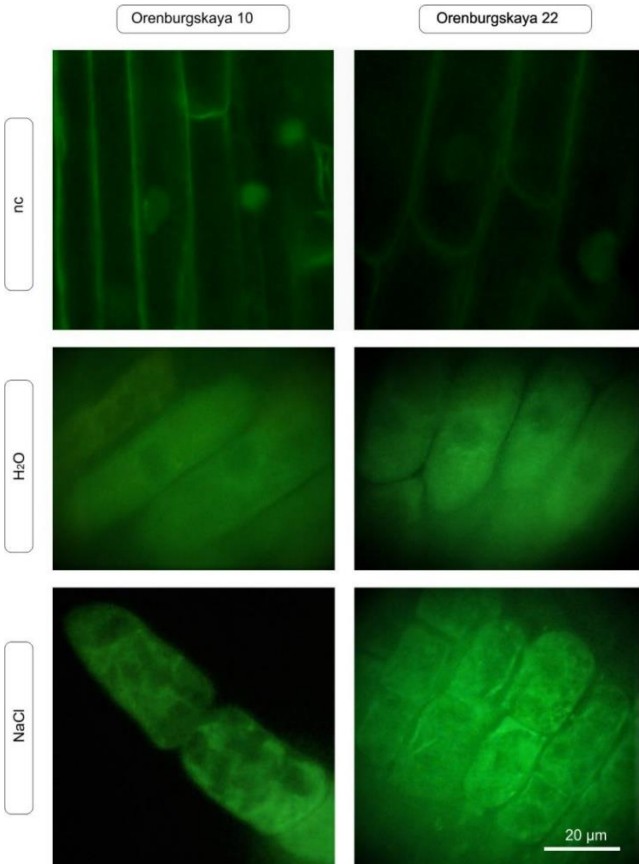

**Figure 6.** In vivo detection of mitochondria using MitoTracker Green FM (green) in 10-day-old wheat root cells under normal plant cultivation conditions and in the presence of NaCl (nc: negative control).

The qualitative ultrastructural changes in NaCl-treated plants' mitochondria (Figure 7c,d) as compared to control plants' mitochondria included a change in the shape of the organelles from round (Figure 7a,c) to oval (Figure 7b,d) with changes in size, matrix optical density and the emergence of cristae with a thickened membrane in both genotypes (Figure 7b,d). Also, electron micrographs showed a change in the electron density of the mitochondria matrix of NaCl-treated sensitive plant cells (Figure 7b).

Staining of the roots of the Orenburgskaya 22 and Orenburgskaya 10 wheat varieties with a fluorescent dye on ROS showed that when there is salinity, ROS is detected in all root tissues (in the control, single stained cells were detected on the root surface); however, the staining intensity varies in cells from different zones. Assessment of the state of the mitochondrial network using MitoTracker Green showed that with an increase in ROS levels in the cells, mitochondria accumulate fluorochrome, which indicates the viability of cells and tissues under these conditions. Since not all root zones were equally stained with ROS, we estimated the distribution of cells with an increased level of ROS in different zones of the roots by combining images taken using phase contrast and fluorescence microscopy (Figure 8).

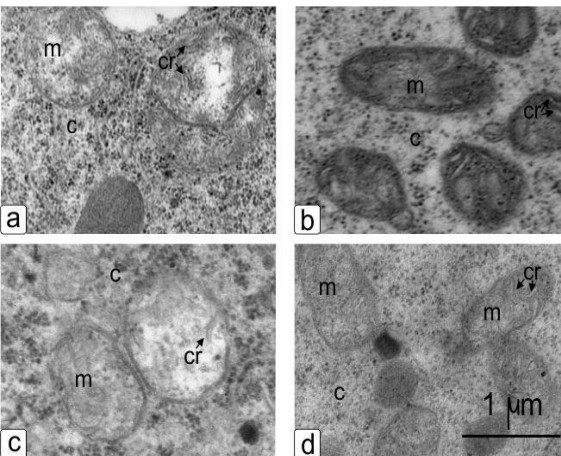

**Figure 7.** Mitochondria in the young cells of the cortex of the main root of wheat during germination in water (**a**,**c**) and in the presence of NaCl (**b**,**d**) after 10 days of germination. Sensitive genotype–Orenburgskaya 10 (**a**,**b**) and resistant genotype–Orenburgskaya 22 (**c**,**d**). Visible changes in shape, cristae and the compaction of the matrix. Cr: crista, m: mitochondria, c: cytoplasma.

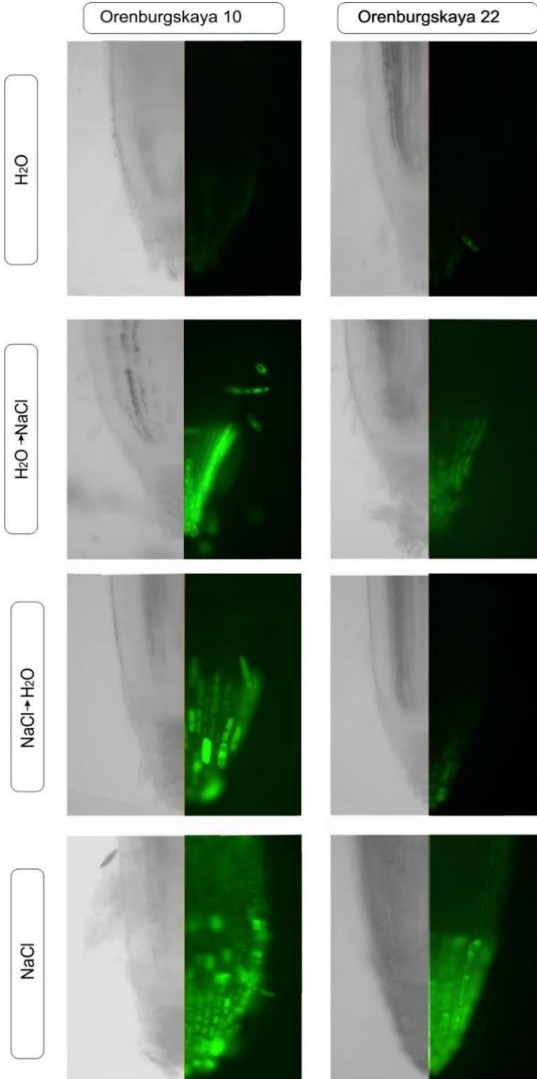

**Figure 8.** Distribution of reactive oxygen species (ROS) + and ROS - cells in the zones of 10-day-old wheat roots.

Then, the root zone with the brightest fluorescence (+++), average level of fluorescence (++), low level (+), and lack of luminescence (-) were noted. The data obtained are shown in Tables 1 and 2, and they show that during salinization, the most intense ROS staining can be observed in the cap and division zone. At the same time, in comparison with the control, an increase in the ROS level is most noticeable in the cells of the epidermis and cortex, and to a lesser extent in the zone of the central cylinder. Therefore, to study the effect of oxidative stress induced by salinization, epidermal and cortical cells from the cap and division zones are preferred.

**Table 1.** Distribution of wheat cells from the division zone to the stretching zone stained with ROS.

| Variant | Cap | Division Zone | Stretch Zone |
|---|---|---|---|
| Orenburgskaya - 22 $H_2O$ | + | - | - |
| NaCl | ++ | ++ | + |
| $H_2O$ + NaCl | ++ | ++ | - |
| NaCl + $H_2O$ | + | + | - |
| Orenburgskaya - 10 $H_2O$ | + | + | - |
| NaCl | +++ | +++ | ++ |
| $H_2O$ + NaCl | +++ | +++ | - |
| NaCl + $H_2O$ | +++ | +++ | ++ |

**Table 2.** Distribution of cells on wheat root tissues stained with ROS.

| Variant | Epidermis | Barc | Central Cylinder |
|---|---|---|---|
| Orenburgskaya-22 $H_2O$ | + | - | - |
| NaCl | ++ | ++ | - |
| H2O + NaCl | ++ | ++ | - |
| NaCl + $H_2O$ | + | - | - |
| Orenburgskaya-10 $H_2O$ | + | - | - |
| NaCl | +++ | +++ | ++ |
| H2O + NaCl | +++ | +++ | - |
| NaCl + $H_2O$ | +++ | +++ | - |

With the cultivation of wheat in the constant presence of 150 mM NaCl, the level of *MnSOD* expression doubles in the leaves of the Orenburgskaya 10 variety, and decreases by 2 times in the roots. In the Orenburgskaya 22 variety, *MnSOD* expression remains unchanged in the leaves, and has a high expression level in the roots compared to leaves. It is interesting to note that in the roots of Orenburgskaya 22 wheat grown in water, the expression level is even higher than in the presence of 150 mM NaCl (Figure. 8). A high level of *MnSOD* expression in common wheat roots may indicate high protective properties against ROS. When changing wheat growing conditions, i.e., when transferring to other media, the level of *MnSOD* expression increases 3-4 times in both varieties of wheat and in both the roots and the shoots of plants (Figure 9).

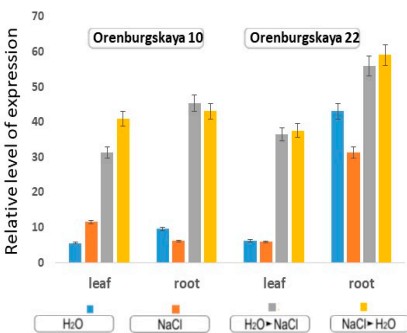

**Figure 9.** Expression gene *MnSOD* in the leaves and roots of 10-day-old wheat based on the mean values and their standard deviations.

This fact may indicate the high adaptive properties of these varieties of wheat to chloride salinity. The primers for the *MnSOD* gene were amplified with DNA from the roots of Orenburgskaya 22 soft wheat and Orenburgskaya 10 durum wheat. Electrophoretic analysis of the PCR products showed that the size of the main product was approximately 122 bp (Figure 10).

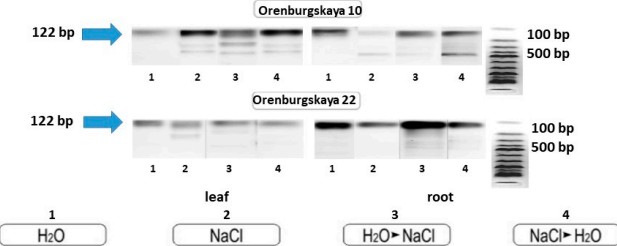

**Figure 10.** PCR amplification of wheat DNA from leaves and roots with primers for the *MnSOD* gene.

## 4. Discussion

The harmful effects of salinization on plant growth and yield are mainly due to two factors: osmotic stress and ionic toxicity [22]. In normal soil, the water potential in the root cells is lower than in the external environment, and the influx of water into the roots occurs through the proteins of the water channel, the so-called aquaporins [23]. In a salty environment, the difference in water potential between the soil and root cells is much smaller than under normal conditions, which leads to a decrease in absorption [24]. As a result, growth inhibition occurs, and ultimately, serious damage to plant tissues. When the internal water balance is disturbed due to abiotic stresses, such as salt stress, plants synthesize and accumulate organic compounds such as polyols, non-reducing sugars, and amino acids [25].

There are two terms related to the response of a plant to abiotic effects, in particular to high salt concentrations: plant tolerance and sensitivity. The first term defines the ability of the plant to mitigate constant exposure to high concentrations of sodium chloride, and the second, to short-term exposure. The proposed scheme for the effect of sodium chloride on the development of wheat allows us to determine not only the tolerance of the plant to the constant effects of salinization, but also the resistance of the plant to salt stress, that is, the short-term effects of a high concentration of NaCl. Salinity induced by NaCl causes a number of specific reactions in germinating seeds. Violations can inhibit swelling, and at later stages, can cause a change in the growth of the primary root, lateral roots, coleoptile or shoot (Figures 2 and 3). The reaction of each plant variety to salinization has individual varietal characteristics. One of the visible symptoms of the effects of salinization is a violation of the growth of seedlings. Despite the varying degree of exposure, the result in all cases will be inhibition of root and/or shoot growth, and the degree of the manifestation of these indicators can be used to estimate the effect of salt toxicity on plant development. Long-term exposure to a high concentration of sodium chloride (150 mM) in durum (Orenburgskaya 10) and soft (Orenburgskaya 22) wheat varieties

caused a decrease in the growth of the aerial part of the plant: by almost 2 times in durum, and only by 20% in soft varieties compared to controls (Figure 3). When wheat is transferred from a salt medium to an aqueous one, the length of the roots decreased in durum wheat by ~60%, and in soft wheat only by ~30%. Morphometric changes occur in both varieties of wheat and have the same tendency. However, the level of reduction in the shoots of the Orenburgskaya 10 variety when transferred from salt to water exceeds the values of the decrease in the shoots of Orenburgskaya 22. From the data obtained it follows that, according to morphometric indicators, durum and soft wheat have a high tolerance to salinization, the same sensitivity to salt stress, but different sensitivity to "water stress". Toxicity from a high concentration of salt can lead to significant damage to plant tissues, and even, the death of the whole plant. Cereal seedlings are a unique and very convenient model for studying PCD (programmed cell death) in plants. First, their growth and development can be easily synchronized [26], which is important when studying the dynamics of many biochemical processes, and secondly, their individual organs are subject to organoptosis [27]. The cereal coleoptile is the first germinal (colorless, green or reddish) leaf of cereals that does not have a leaf blade and is a closed tube. It has protective properties for subsequent leaves. So, the coleoptile of cereal functions for a relatively short time and quickly dies as the seedling forms and grows (Figure 4). Despite the fact that PCD is rather rigidly programmed in the plant ontogenesis, it can be induced or modulated by various environmental factors or agents, including various infections and stressful effects of an abiogenic nature (hypoxia, oxygen stress, and others) [28].

The first problem in the analysis of PCD in cell cultures is the quantification of cell viability/death before long [29]. A change in the growth rate also indicates a violation of the water metabolism and respiration due to a violation of transpiration and the growth process by stretching (Figure 5). Breathing sharply increases during stress, and the production of superoxide radicals is strictly related to the frequency of mitochondrial respiration [30]. During the germination of durum and soft wheat seeds, a gradual increase in respiration occurs, which provides energy by processing the reserve polysaccharides, mainly starch of endosperm. After the depletion of these reserves and the plant enters the phase of active photosynthesis, provided by the appearance of the first photosynthetic leaves, energy is provided by the use of newly synthesized sugars by mitochondria. The seedlings of the studied wheat samples showed no significant differences in respiration (Figure 5). The presence of sodium chloride caused an acute increase in respiration, which was more clearly expressed in durum wheat. When the growing conditions are changed by increased salinization, a decrease in respiration intensity occurs, although this was only significant in the case of durum wheat. When transferring plants cultured in sodium chloride to water, there is also a significant increase in respiration. It can be assumed that this is caused by a significant increase in growth and the formation of new leaves.

Mitochondria are one of the main organelles generating ROS and regulating the redox potential of the cell [31]. It has been established that changes in mitochondria affect (Figure 6) various aspects of cell adaptation and death through the release of a number of mitochondrial proteins, change in the transmembrane potential difference, formation of reactive oxygen and nitrogen forms, disruption of the electron transport chain, and inhibition of ATF synthesis [32]. The intensification of lipid peroxidation in mitochondrial membranes leads to a violation of the integrity of the membranes, swelling, and subsequent lysis of mitochondria. Damage to the structure of mitochondria, which are the main energy-generating organelles, under conditions of oxidative stress can lead to disruption in the energy supply of cells necessary for adaptation in stressful conditions.

An important functional test for oxidative stress is the intravital staining of mitochondria using specific fluorescent markers of the MitoTracker family (Figure 6).

The ultrastructural changes in mitochondria (Figure 7) include changes in the shape, matrix clarification and form of cristae, or conversely, contraction of the matrix density and the emergence of many cristae with a visible geometric profile and a less discernible, thickened membrane [33]. In salt-treated plants as compared to control plants, mitochondria were observed to be deficient in the presence of altered cristae, enlarged and the matrix appeared pale [34]. Electron micrographs

showed changes in the electron optical contrast in the mitochondria matrix [35,36]. All of the observed structural responses included changes in density, size, and shape, which indicates an osmotic effect, which was probably not directly related to the action of ions but was caused by an increased response in osmolytes, and is characteristic of plants' defense reactions to stressor effects [37] or oxidative damage and impaired sugar transport [38].

The amount of ROS increases during chemical and environmental stresses, including cold, drought, floods, treatment with herbicides, attack of pathogens and ionizing radiation [39]. ROS are currently considered as signaling molecules, their role is shown in stress responses to the implementation of proliferation and differentiation programs, natural and induced aging, and plant cell death.

One of the common criteria for assessing the oxidative status of plant cells is the detection of ROS using the Carboxy reagent, H2D2FFDA (Figure 8). It is known that ROS perform both signal and regulatory functions in plant cells during salinization [40]. They are formed in various cell compartments: in chloroplasts, mitochondria, peroxisomes, the plasma membrane, cytosol, and the cell membrane [41]. NaCl is often considered extremely toxic to plants. The changed status of water, the imbalance of ions and hyperosmotic stress caused by NaCl treatment induce further growth inhibition and molecular damage during the formation of reactive oxygen species [22]. Assessment of the mitochondrial network using MitoTracker Green on both wheat genotypes showed that mitochondria accumulate fluorochrome with an increase in ROS levels in the cells, which indicates the viability of cells and tissues under these conditions (Figure 8). We established an increase of ROS level in cells of durum wheat.

In response to a high concentration of salts, various genes begin to regulate molecules that directly or indirectly participate in plant protection [42]. Some genes encoding osmolytes, ion channels, receptors, and some other regulatory signaling factors or enzymes are capable of imparting salt tolerance to phenotypes when transferred to sensitive plants. Susceptibility or tolerance to stress under the influence of a high concentration of sodium chloride in plants is a coordinated action of many genes that respond to stress [25]. Cellular toxicity caused by a high content of Na + ions is the predominant ionic toxicity; it leads to inhibition of various processes, such as absorption of K +, inactivation of vital enzymes, and inhibition of photosynthesis. The resistance of plants to salinization is due to the presence of specific or nonspecific mechanisms to ensure stable metabolism, growth and development in the ontogenesis of plants associated with sensitivity to one or more types of stress factors, namely, the osmotic, oxidative or toxic stress effects of NaCl [43].

The changed status of water, the imbalance of ions, and hyperosmotic stress caused by exposure to a high ionic concentration of NaCl cause the formation of ROS, which leads to a further slowdown in plant growth and its death [28,29]. Various mechanisms exist to reduce the negative effects of ROS in plants. One mechanism is the enzymatic reduction of ROS toxicity. In this study, we examined the expression of superoxide dismutase (*MnSOD*) genes, which is required in metabolism [11].

With the cultivation of wheat in the constant presence on 150 mM NaCl, the level of *MnSOD* expression in the leaves of the Orenburgskaya 10 variety doubles, but in the roots it decreases by 2 times. In the Orenburgskaya 22 variety, *MnSOD* expression remains unchanged in the leaves, and *MnSOD* has a high expression level in the roots compared to with leaves. When we changed the wheat growing conditions, i.e., when plants were transferred to other media (from water to salt, and from salt to water), the level of *MnSOD* expression increased 3–4 times in both varieties of wheat and in both the roots and shoots of plants (Figure 9).

The primers for the *MnSOD* gene were amplified with DNA from the roots of Orenburgskaya 22 soft wheat and Orenburgskaya 10 durum wheat. PCR amplification of products showed that the size of the main product was approximately 122 bp on both genotypes (Figure 10). However, in wheat samples, especially Orenburgskaya 10 cultivar grown with a long presence of salt, as well as with short-term exposure to salt, additional transcripts are formed compared to the control variants. Since superoxide dismutase isoform sequences have homologous sequences, it can be assumed that the salt increases the availability of other enzyme isoforms. Moreover, additional bands of PCR products are

possibly formed as a result of cytosine methylation of DNA in wheat grown in the presence of sodium chloride [44]. The number of additional PCR products depends on the variety of wheat, and on its sensitivity to high concentrations of sodium chloride. The appearance of additional PCR products is probably associated with an increase in the protective properties of ROS. According to biometric data, the root system of the Orenburgskaya 10 variety is more developed and less susceptible to the negative effects of chloride salinity than the Orenburgskaya 22 variety. The expression of other *MnSOD* isoforms is probably what stimulates the adaptation of durum wheat to NaCl.

## 5. Conclusions

Wheat is one of the most important crops in the world, accounting for more than half of total human food consumption. However, its safe production is seriously threatened by natural disasters such as drought, salinization and extreme temperatures, which lead to a significant decrease in annual yield and wheat quality. Accordingly, increasing the resistance of wheat to salt stress is one of the most important tasks of the breeding program. Comparison of wheat genotypes with different resistance to chloride salinity according to the functional state of cells using intravital markers could be effectively used in the future for the rapid assessment of the oxidative status of plant root cells grown under the influence of various stress factors. In the course of the study, experimental data were obtained to identify stable genotypes that allowed us to characterize the systemic cellular response to stress effects of an abiotic nature (using the action of edaphic stress as the toxic effect of sodium chloride) in the cells of the roots of wheat seedlings. New cytological criteria have been developed to determine, at the early stages of development (seedlings), the functional state of cells in whole plant roots, which will make it possible to assess the manifestation and stress parameters: stability/destruction of the structural organization of root cells, and preservation/destruction of coleoptile cells. *SOD* gene analysis can provide important information for genetic improvement in the resistance of soft and hard wheat.

Thus, from the data obtained it follows that at first glance (according to morphometric indicators), Orenburgskaya 10 and Orenburgskaya 22 varieties have the same resistance to salt stress. However, a more detailed study reveals differences in wheat varieties in the accumulation of ROS. These differences are most likely related to the difference in the mechanisms of neutralization and elimination of ROS from the roots and the shoots of wheat seedlings of different varieties. This is due not only to the high level of *MnSOD* expression, but also to the expression of their isoforms. Due to these differences, we can conclude that the Orenburgskaya 22 variety is more resistant than Orenburgskaya 10. Thus, such an integrated approach to studying salt tolerance is necessary to characterize the resistance of wheat varieties to salinization.

**Author Contributions:** N.K. performed light and fluorescent microscopy, evaluated data, wrote and finalized the manuscript; E.B. performed electron microscopy, evaluated data; E.A. performed assessment and measurement respiration intensity; T.D. performed the experiment, and obtained and characterized plants; L.F. designed and performed the experiment, P.C.R, designed, and prepared figures, evaluated data, wrote and finalized the manuscript. All authors have read and agreed to the published version of the manuscript.

**Funding:** The reported study was supported by project RFBR 18-016-00150 and partially by assignment 0574-2019-002 of the Ministry of Science and Higher Education of the Russian Federation.

**Acknowledgments:** We are grateful to Galina Baranova for technical support.

**Conflicts of Interest:** The authors declare no conflict of interest.

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
