# Peer review of "Oxidative Damage to Various Root and Shoot Tissues of Durum and Soft Wheat Seedlings during Salinity"

_agriculture, doi:10.3390/agriculture10030055_

Round 1
Reviewer 1 Report
The manuscript entitled “Oxidative Damage to Various Root Tissues and Aerial Parts of Durum and Soft Wheat Seedlings during Chloride Salinity” by Kononenko et al. compared the salinity stress response between bread and durum wheat. They concluded that bread wheat showed significant less oxidative damage in roots than the durum wheat. They used fluorescence microscopy and SEM imaging techniques to study the root responses between bread and durum wheat under salt stress.
Some comments:
Only one bread wheat and one durum wheat were used. There are many sensitive wheat varieties and relative tolerant varieties. How about the variation within the relative tolerant wheat varieties and sensitive varieties? In the title, chloride salinity was used. However, no comparison of different chloride salt was tested in this study. Author need to clarify why it is related to chloride salinity but not NaCl salinity. Or consider to change the words. The quality of the microscopy images needs to be improved. Many nonspecific bands were showed in Figure 10. Please check if the designed primers are appropriate. The more in-depth discussion is needed. Some data can be moved into supplementary, e.g. Figure 10. Please add the statistic analysis for Figure 3, 5, and 9. The cited literatures are old. Please cite more literatures published in recent 2 to 3 years. For example, there are many good review papers out in these years.
Reviewer 2 Report
Review to the manuscript: Oxidative Damage to Various Root Tissues and Aerial Parts of Durum and Soft Wheat Seedlings during Chloride Salinity (Agriculture 694534)
The manuscript follows selected parameters of wheat plants subjected to salinity stress (plant growth, oxidative damage, expression of MnSOD gene). In spite of promising abstract indicating the identification of cytological and molecular markers suitable for selection of salinity-resistant cultivars, I feel that the manuscript cannot be published in its present form. See detailed comments below.
Methods are unclear and incomplete. The salinity level (NaCl concentration) used for experiments isn´t mentioned (this information is at the end of discussion only). The composition of cultivation solution is not clear (distilled or tap water? no nutrient added?). Electron microscopy samples preparation is not mentioned. The estimation of mitochondria size is not described (number of biological repetitions, the number of organelles measured, any image analysis involved?). The reason for DNA extraction and amplification is not clear. The software for primer design and qPCR quantification is not shown, etc. The data presentation in the Results section is also far from being acceptable. The statistical evaluation is missing (no statistical test is presented). The numbers of repetitions are not clear. Is STD or SE presented in graphs? What are numbers 1 and 2 in fig. 3 and 5 (genotypes are not indicated)? Negative controls for staining procedures are missing (especially in fig.6). The quality of microphotographs in Fig.4 is very low. At least white balance should be adjusted to improve the visibility of trypan blue staining. I would also prefer to change “intravital” staining to viability test for trypan blue staining. If I understand well, data presented in fig.10 are the results of DNA (not cDNA) amplification. Hence, the products of different sizes hardly be additional transcripts. These are rather unspecific amplification products (annealing temperature of PCR is not shown in methods). Discussion is weak. The English has to be corrected by a native speaker. The format of references in the list should be adjusted to fit journal format. Last, I have doubts about the selection of sensitive x resistant cultivars that are compared in the study. The study compares two different species of Triticum (as resistant and sensitive cultivar). This may limit future usage of any identified marker in selection of cultivars within one species.Author Response
Please see attached file.

Reviewer 3 Report
The manuscript needs to be carefully checked for language. The presentation of some sections need to be improved. Please see below for details.
Introduction
Line 34: delete the full-stop in front of number 1
Materials and methods
Line 81: What does it mean by 5001k? usually the light intensity used in experiment is described, and the common unit is lux. Please revise.
Lines 79 – 84: please briefly describe the roll culture method as the reference is in Russian and was published in 1968 therefore the readers cannot easily to get access for more details. Figure 1 is not well presented. Does it mean 7-day-old wheat plants were transferred to NaCl solution for 3 days then back to water? Which concentration of NaCl was used?
Lines 85 – 88: Please provide the supplier of Trypan blue and protocol that was followed (e.g ThermoFisher, abcam …and manufacturer’s instructions).
Line 87, 93 & 97: 3-fold washing; does it mean 3 times washing?
Line 90: The root tip (4-5 mm) of the wheat germ at what age? Before or after exposing to NaCl or both?
Line 91: what does it mean by: of at least 5 pcs?
Line 94: Please specify what are the experimental solutions
Line 106: Please specify Which solution and how many plants in these sentences: Plants were placed in water or solution. Than (then?) some rooted plants …
Line 108: threefold replication; Do the author mean 3 replicates?
Line 120: DNA was isolated from individual organs of wheat…Please specific which part of the plants were used? Leaf, root, stem?
Line 124: concentration of isolated RNA preparations was determined spectrophotometrically. Please specify the equipment used (i.e NanoDrop ….)
Line 132: RT-PCR
Line 133: What is PCR-RV?
Line 135: Please correct the writing form of temperature
Result section
Figure 2: Please provide high resolution photos.
The labels in the photos are not clear. Please present the labels in a better way.
The legend of Figure 2 is insufficient details. Seedlings of soft and durum wheat after germination (How many days after germination were the photos taken? E.g 3 days, 7 days?)
Figure 3: The legend is insufficient details. Please provide more details about what time (day of experiment) the data were taken? Data are Mean + standard error or Standard deviation? How many replicates? Any statistic analysis was done?
Figure 4: The legend is insufficient details. Please provide more details about what time (day of experiment) the samples were stained?
Figure 5: The legend is insufficient details. Please provide more details about what time (day of experiment) the data were taken? Data are Mean + standard error or Standard deviation? How many replicates? Any statistic analysis was done? What do number 1 and 2 represent?
Figure 6: The legend is insufficient details. Please provide more details about what time (day of experiment) the samples were stained?
Figure 7: The legend is insufficient details. Please provide more details about what time (day of experiment) the samples were stained? Please label the image (a,b,c,d …) and explain in the legend.
Figure 10: proper label for each lane of the gel and appropriate molecular ladder are needed. Legend needs to be improved with more details e.g which lane contains which sample, the time (day of experiment) that samples were collected for DNA extraction
Discussion
the discussion is not cohesive and need to be re-witten
Round 2
Reviewer 1 Report
The authors have successfully addressed most of my comments.
Author Response
Thank you for your valuable comments. Manuscript checked by a native speaker. Edits marked in yellow have been added to the text
Reviewer 2 Report
Revised version of the manuscript 694534:
The authors made significant improvement in some parts of the manuscript. The method section was considerably improved. Missing points were added. There are however still many weak points in other parts of the manuscript that have to be solved before publication.
The result section and the discussion are still weak in many aspects. Statistics is unclear. One level of significance in the figure capture is not enough (especially in figures with more than one graph and four treatments). What does it mean? Some post-hoc test is needed to indicate significant differences among individual treatments. Fig. 9 misses the statistics at all.
The quality of some pictures isn´t satisfactory, especially Fig. 4. I also still doubt about the interpretation of the data presented in Fig. 10. Fig. 7: Authors did not perform any quantitative analysis of mitochondria. The differences in their size and shape, which are presented in the result section, are therefore questionable.
In spite of the fact that authors declare the language correction by relevant authorities, typographic errors are present (e.g. genotipes in line 390).
In summary, I think that the manuscript still needs to be improved before publication.
Author Response
Thank you for your valuable comments. Manuscript checked by a native speaker. Edits marked in yellow have been added to the text.
We took into account your comments and made additions to the results and discussions.
Methods of statistical analysis are given in the Methods section. The program Statistica 5.0, Student’s parametric criteria, and standard Microsoft Excel software were used for date evaluation. The arithmetic mean values were calculated by the formula: Mx= Σ Xi/n. Standard deviations: σÑ…=ÖDx= ÖΣ(Xi–Mx)2 / n-1, where : DÑ… is the dispersion.
Figure.9. The mean values and their standard deviations were made.
Figure.4. Unfortunately, the coleoptile samples were not monolayer, so the images were obtained with insufficient resolution.
Figure.7. We used a qualitative assessment. We wanted to show what mitochondria look like on an ultrathin section of different genotypes during salinization.
Fig.10. We agree that this is a complicated and controversial question. Salt leads to a change in chromatin structure, which may be accompanied by a change in genome activation. It is known that wheat is a polyploid, and the activation of genes in it is complex. We gave a general picture of PCR amplification for discussion with interested colleagues and for developing directions for solving this problem.
We tried to correct technical errors.
Reviewer 3 Report
Materials and methods
Lines 81 – 84: Please specify whether more solution (distilled water or 150 mM NaCl) was added to glasses after ½ rolls were moved to. If yes, how much solution was added?
Line 85-86: According to information described in lines 78-79, each paper roll contained 50 seeds in each replicate. After 7 days, ½ roll was moved to other condition either NaCl or water, i.e 25 seedings were moved to other condition, how 30 seedlings were obtained (n=30, in line 85) for statistic? Please explain.
Line 89: Paper rolls were cultivated in distilled water and 150 mM NaCl. This sentence does not make sense. It should be re-written, e.g: Seeds of Orenburgskaya10 and Orenburgskaya 22 on paper rolls were cultivated in distilled water or 150 mM NaCl …
Lines 93-95: The comments from previous review version has not been addressed. Please provide the supplier of Trypan blue and protocol that was followed (e.g ThermoFisher, abcam …and manufacturer’s instructions).
Line 147: in publications the term RT-PCR was used for Real-time PCR so please change PCR-RT to RT-PCR in this manuscript to be consistent with other scientific publications.
Result section
All figures legends should be carefully check with sentence structure and meaning.
Author Response
Thank you for your valuable comments. Manuscript checked by a native speaker. Edits marked in yellow have been added to the text
The seeds of wheat (50 pieces each) were laid out on paper rolls. Paper rolls (4 pieces each) were placed in glasses. The experiment was carried out in 3 replicates (3 glasses each). In the experiment, 12 glasses, 48 rolls and 600 seeds were taken for each wheat variety. In 6 glasses of them 150 ml of water was added, and in the other 6 - 150 ml of 150 mM salt. After 7 days, the solutions were replaced with new ones, 150 ml each. In 3 glasses, which were with water, replaced with water, and in the other 3 glasses, water was replaced with 150 mM salt. In 3 glasses with 150 mM salt, the solution was replaced with new, and in the other 3 glasses, with 150 mM salt was replaced with water.
After 10 days, for statistics, 30 seedlings were selected from each roll.
Staining with trypan blue (AppliChem GmbH, Germany) - 0.5% aqueous solution was carried out for 5 min with 3-time washing. The dye used for selective staining of dead cells and tissues does not stain living cells. Designed for research purposes.